

# Integrated transcriptomic and proteomic analysis of *Tritipyrum* provides insights into the molecular basis of salt tolerance

Rui Yang[1,*], Zhifen Yang[1,*], Ze Peng[1], Fang He[1], Luxi Shi[1], Yabing Dong[1], Mingjian Ren[1,2], Qingqin Zhang[1], Guangdong Geng[1] and Suqin Zhang[1,2]

[1] Guizhou University, Guiyang, China
[2] Guizhou Subcenter of National Wheat Improvement Center, Guiyang, China
* These authors contributed equally to this work.

## ABSTRACT

**Background:** Soil salinity is a major environmental stress that restricts crop growth and yield.

**Methods:** Here, crucial proteins and biological pathways were investigated under salt-stress and recovery conditions in *Tritipyrum* 'Y1805' using the data-independent acquisition proteomics techniques to explore its salt-tolerance mechanism.

**Results:** In total, 44 and 102 differentially expressed proteins (DEPs) were identified in 'Y1805' under salt-stress and recovery conditions, respectively. A proteome-transcriptome-associated analysis revealed that the expression patterns of 13 and 25 DEPs were the same under salt-stress and recovery conditions, respectively. 'Response to stimulus', 'antioxidant activity', 'carbohydrate metabolism', 'amino acid metabolism', 'signal transduction', 'transport and catabolism' and 'biosynthesis of other secondary metabolites' were present under both conditions in 'Y1805'. In addition, 'energy metabolism' and 'lipid metabolism' were recovery-specific pathways, while 'antioxidant activity', and 'molecular function regulator' under salt-stress conditions, and 'virion' and 'virion part' during recovery, were 'Y1805'-specific compared with the salt-sensitive wheat 'Chinese Spring'. 'Y1805' contained eight specific DEPs related to salt-stress responses. The strong salt tolerance of 'Y1805' could be attributed to the strengthened cell walls, reactive oxygen species scavenging, osmoregulation, phytohormone regulation, transient growth arrest, enhanced respiration, transcriptional regulation and error information processing. These data will facilitate an understanding of the molecular mechanisms of salt tolerance and aid in the breeding of salt-tolerant wheat.

## INTRODUCTION

Wheat (*Triticum aestivum*) is the most important food crop in the world. Its total area, total output and total trade volume all rank first among food crops, and it provides 19% of available calories for humans (*Yan et al., 2020*). It is predicted that by the middle of the 21st century, increasing salinization will lead to the loss of 50% of the world's arable land

Corresponding authors
Guangdong Geng,
gdgeng@gzu.edu.cn
Suqin Zhang, sqzhang1@gzu.edu.cn

(*Munns & Tester, 2008*; *Goyal et al., 2016*). Salt stress is a major challenge for wheat production because it has devastating effects on crop yield. Salt stress can occur at any time during the life cycle of wheat (*Setter et al., 2016*; *Ghobadi, Ghobadi & Zebarjadi, 2017*). Therefore, finding ways to improve crop tolerance to salt stress is essential to improve wheat productivity and achieve food security.

Salt accumulation in the soil causes osmotic imbalance, ion toxicity, nutritional disorders, normal metabolic disruption, reactive oxygen species (ROS) accumulation and oxidative stress-related injury (*Tang et al., 2015*; *Ma et al., 2020*). Plants have evolved complex cellular, physiological and molecular mechanisms for responding and adapting to environmental cues, including salt stress (*Chinnusamy, Jagendorf & Zhu, 2005*; *Deinlein et al., 2014*; *Fahad et al., 2015*; *Kumar et al., 2017*; *Mahajan et al., 2017*; *Mahajan et al., 2020*). Physiological studies have demonstrated that osmotic adjustment, ion homeostasis, antioxidant systems, hormonal regulation, nutrient uptake and signaling play important roles in salt tolerance (*Zhao et al., 2020*). In addition, dramatic alterations in gene expression, which lead to changes in protein profiles, are closely related to these processes (*Du et al., 2010*; *Sobhanian, Aghaei & Komatsu, 2011*).

Proteins are directly involved in shaping plant phenotypes through either structural or regulatory functions related to the formation of the plant epigenome, transcriptome and metabolome. Quantitative proteomics has become a powerful tool in the identification of proteins and mechanisms involved in salt responses and tolerance. The enhanced salinity tolerance of durum wheat appears to be governed by a higher capacity for osmotic homeostasis, a more efficient defense by late embryogenesis abundant (LEA), redox and pathogenesis-related proteins, and an improved protection from mechanical stress by increased cell wall lignification (*Capriotti et al., 2014*). The enhanced salinity tolerance of the salt-tolerant barley 'DH187' results mainly from the increased activity levels of signal-transduction mechanisms, which eventually leads to the accumulation of stress-protective proteins and cell-wall structural changes (*Mostek et al., 2015*). A total of 514 and 770 protein spots were reported in the most salt-tolerant and salt-sensitive rice cultivars, respectively. The differentially expressed proteins (DEPs) are associated with major metabolic pathways, including photosynthesis, energy metabolism, amino acid metabolism and nitrogen assimilation, as well as stress and signaling pathways (*Frukh et al., 2020*). In salt-stressed maize '8723', the DEPs are primarily associated with phenylpropanoid biosynthesis, starch and sucrose metabolism, and mitogen-activated protein kinase signaling pathways (*Chen et al., 2019*). It is important to identify valuable key genes and proteins from the large amounts of data obtained by transcriptome sequencing and proteome technologies. In total, the expression levels of 13 and 11 genes in salt-tolerant and -sensitive barley genotypes are consistent after salt stress at the transcriptome and proteome levels, respectively (*Lai et al., 2020*). In salt-tolerant sweet potato, eight differentially expressed genes (DEGs) display the same pattern under salt-stress conditions at the transcriptome and proteome levels (*Meng et al., 2020*).

SWATH-MS is a recently introduced technique consisting of a data-independent acquisition (DIA) and a targeted data analysis strategy that aims to maintain the favorable

quantitative characteristics (accuracy, sensitivity and selectivity) achieved in targeted proteomics but on the scale of thousands of proteins. The E genome species (*e.g.*, halophile wheatgrass *Thinopyrum elongatum*) of the *Triticeae* are invaluable sources of salt tolerance (*Jauhar, 1990*; *Munns & Tester, 2008*; *Margiotta et al., 2020*). *Tritipyrum* derived from the wide crosses of *Triticum* and *Thinopyrum* show salt tolerance (*King et al., 1997*; *Yuan & Tomita, 2015*). Currently, there are limited reports on the molecular mechanisms of *Tritipyrum* salt tolerance; therefore, they remain largely unknown. Here, protein markers and pathways were identified using DIA proteomic and transcriptomic profiling, and these will aid in elucidating the molecular mechanisms involved in the salt tolerance of *Tritipyrum*. The results provide valuable information for the molecular breeding of improved salt tolerance in wheat and other crops.

## MATERIALS AND METHODS

### Plant materials

'Y1805' is a stable progeny from the wide cross between *Triticum aestivum* and *Thinopyrum elongtum*. Salt-tolerant *Tritipyrum* ('Y1805') and salt-sensitive *Triticum aestivum* ('Chinese Spring', 'CS') were used in this experiment. These genotypes were chosen on the basis of preliminary experiments. 'Y1805' shows plump kernels, mid-maturity and high complex disease-resistance levels, including immunity and high resistance levels to scab, rust and powdery mildew of wheat.

### Plant growth conditions and stress treatments

The seeds of 'Y1805' and 'CS' were germinated in a growth chamber (relative humidity of 75% and a 20/15 °C light/dark photocycle). The seedlings were sown on a floater board in 1/2 Hoagland's solution with a 16-/8-h light/dark cycle, irradiance of 400 μmol m$^{-2}$s$^{-1}$ and the same temperature and humidity as in the germination chamber. The culture solution was refreshed every 3 days. On the 14$^{th}$ day (two-leaf stage), salt-stress treatments (1/2 Hoagland's solution supplemented with 250 mM NaCl) were started. A 250-mM NaCl concentration was used as a result of preliminary gradient tests. The pH of the culture solution was 6.0. The first wheat root and shoot samples of a uniform size were selected at 5 h (T1 stage) after exposure to salt stress. The materials were recovered (in 1/2 Hoagland's solution without NaCl) after 24 h of salt stress. The second and third samplings were performed at 1 h (T2 stage) and 24 h (T3 stage) after recovery, respectively. These sampling stages were chosen on the basis of the phenotypic and physiological data. Normal (1/2 Hoagland's solution without NaCl) control cultured materials, CK1, CK2 and CK3, were used as parallel controls for T1, T2 and T3, respectively. The root samples of the T1 and T2 stages were immediately frozen in liquid nitrogen after sampling and stored at −80 °C for the transcriptomic analysis, proteomic analysis, and confirmational qRT-PCR. The samples at the T3 stage were only used for growth analysis. Three biological replications and two technical replications were employed, and at least 10 seedlings were mixed per replicate.

## Protein analysis

### Protein extraction and DIA-MS/MS analysis

Total proteins of wheat roots were extracted using the phenol method with slight modifications (*Isaacson et al., 2006*). Briefly, 0.5 g of wheat roots were ground into a fine powder in a lysis buffer containing 8 M urea, 400 mM ammonium bicarbonate, 10 mM dithiothreitol and a protease inhibitor cocktail (Roche, Basel, Switzerland). Then, 2× volume of phenol saturated with Tris-HCl (pH 7.5) was added, and centrifuged at 25,000 g for 15 min at 4 °C. The upper phenolic phase was collected, and then, 5× volume of precooled precipitation solution, containing 0.1 M ammonium acetate in methanol and 10 mM dithiothreitol, was added to the protein mixture. Each sample was maintained for 2 h at −20 °C. Then, the samples were centrifuged at 25,000 g for 15 min at 4 °C, and the supernatant was removed. The pellets were further washed with precooled acetone with centrifugation at 20,400 g for 40 min. Then, trypsin [protein:trypsin = 40:1 (w/w)] was added for enzymolysis, followed by desalination, vacuum drying, and redissolution. Protein concentrations were measured by using Nanodrop ND-1000 (Thermo Fisher Scientific, San Jose, CA, USA).

The peptide samples were separated using an UltiMate 3000 UHPLC (Thermo Fisher Scientific, San Jose, CA, USA), ionized by a nanoESI source and then put into a Q-Ex Active HF tandem mass spectrometer (Thermo Fisher Scientific, San Jose, CA, USA) for DIA mode detection. The spectral library was constructed using the DIA of targeted samples. Spectronaut was used to effectively deconvolute, accurately identify and quantitatively analyze the data (*Bernhardt et al., 2014*).

### DEP identification and bioinformatics analysis

The peak area of an ion pair was extracted using Spectronaut. Then, the Msstats software package was used to complete the error correction and normalization steps (*Choi et al., 2014*). Two conditions, fold change ≥ 2 and false discovery rate < 0.05, were used as the DEP criteria. The Gene Ontology (GO) and Kyoto Encyclopedia of Genes and Genomes (KEGG) enrichment analyses, protein-protein interaction (PPI) and subcellular localization analysis were performed for these DEPs.

### Proteome-transcriptome-associated analysis

The basis of proteome–transcriptome association analysis is the central principle. Correlation analyses of mRNAs and proteins having the same change trend were carried out, including expression correlation analyses and metabolic pathway map integration analyses.

### The quantitative RT-PCR (qRT-PCR) analysis

A total of 15 candidate proteins were verified by qRT-PCR (Table S1). Total RNA from the same samples used in the proteome analysis were reverse-transcribed using Power SYBR® Green PCR Master Mix (Applied Biosystems, Foster, CA, USA). qRT-PCR amplification was performed on an ABI StepOne Real-Time PCR System. The relative expression levels were calculated in accordance with the $2^{-\Delta\Delta Ct}$ method, with three biological

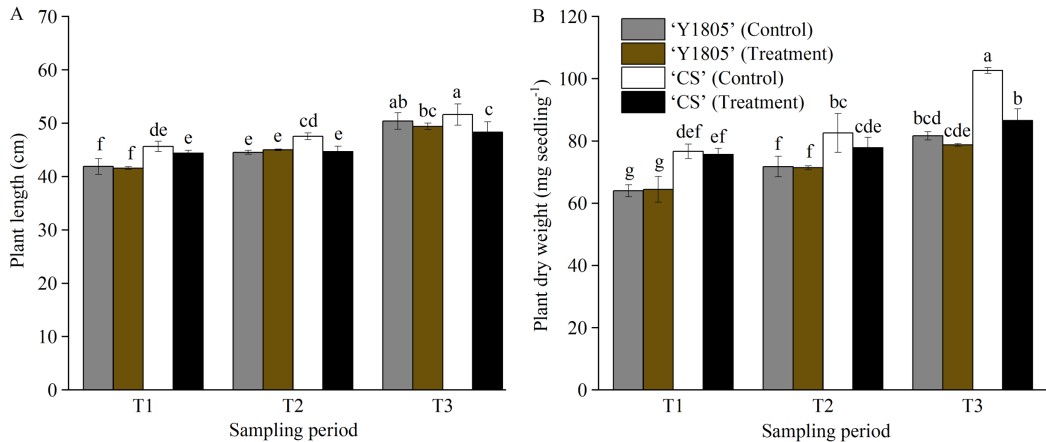

**Figure 1 Effects of salt stress and recovery on the plant length (A) and dry weight (B) in two wheat varieties.** 'Y1805' and 'CS' represent *Tritipyrum* and wheat 'Chinese Spring', respectively. T1, T2 and T3 indicate 5 h after exposure to salt stress, 1 h and 24 h after recovery, respectively. The treatment indicates salt stress at the T1 stage, and recovery at the T2 and T3 stages, respectively, and the control shows its respective corresponding control plant. Bars show means ± SDs (*n* = 3). Values with different letters are significantly different at *p* < 0.05.           

replications and three technical replications (*Depuydt & Hardtke, 2011*), and *β-actin and 18S RNA* were used as the internal control.

## Statistical analyses

The data were analyzed using SPSS statistics 19.0 (IBM Corp., Armonk, NY, USA). Statistical differences between means were determined by Duncan's multiple range test at a significance level of *p* < 0.05 after displaying a significant effect during an ANOVA. Pearson's correlation analysis of binary variables was performed, and two variables were considered significantly correlated at the *p* < 0.05 level.

## RESULTS

### Response of plant growth to salt stress and recovery in two wheat varieties

The plant length and dry weight of salt-tolerant *Tritipyrum* 'Y1805' did not change significantly at any of the three stages compared with the corresponding controls (Fig. 1). Additionally, 'Y1805' maintained normal growth under salt-stress conditions. The plant length of salt-sensitive 'CS' was significantly (*p* < 0.05) less than that of the control at the T2 stage. At the T3 stage, the plant length and dry weight of 'CS' decreased significantly (*p* < 0.05) by 6.37% and 15.69%, respectively, compared with the controls. Thus, 'Y1805' showed a strong salt tolerance under salt-stress conditions and a rapid recovery capability after the salt stress was removed.

### Quantitative analysis of DEPs

The DEP quantitative analysis was carried out under salt-stress and recovery conditions in 'Y1805' and 'CS'. Under salt-stress conditions, 44 DEPs were identified in 'Y1805' ('Y1805' T1), among which 21 were upregulated and 23 downregulated (Fig. 2A). For 'CS'
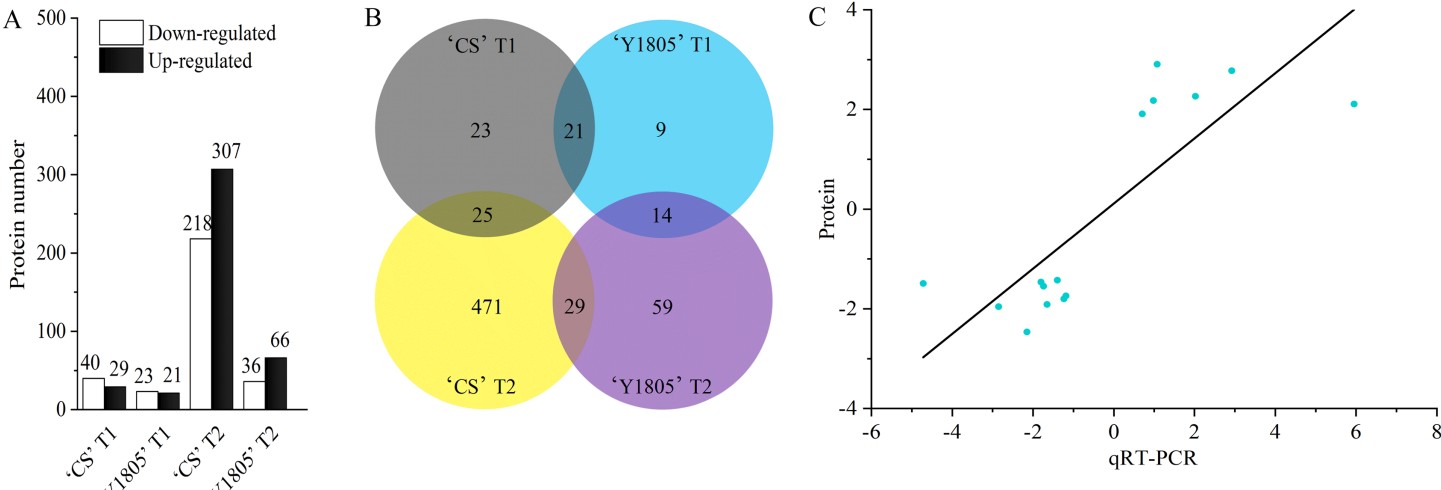

**Figure 2 Differentially expressed protein statistics (A), Venn map (B) and qRT-PCR confirmation (C).** (A, B) CS represents wheat 'Chinese Spring'. T1 and T2 indicate salt stress and recovery, respectively. (C) The x- and y-axes show the relative expression levels ($\log_2$fold change) analyzed independently by qRT-PCR and proteome, respectively.

('CS' T1), 69 (29 upregulated and 40 downregulated) DEPs were identified. Among these DEPs, 21 (10 upregulated and 11 downregulated) proteins were common to both genotypes and showed the same expression patterns (Fig. 2B, Table S2), indicating that they represented the core genes associated with responses to salt stress. 'Y1805' had 23 specific DEPs, including 11 upregulated and 12 downregulated DEPs, which should play key roles in its strong salt-tolerance.

During the recovery process, 102 (66 upregulated and 36 downregulated) DEPs were identified in 'Y1805' ('Y1805' T2). In 'CS' ('CS' T2), 525 DEPs were detected. A total of 29 DEPs appeared in both varieties (Table S3), with 96.55% displaying the same expression patterns. 'Y1805' had 73 (48 upregulated and 25 downregulated) specific DEPs that might contribute to its fast recovery after salt stress.

In total, 14 DEPs were identified under both salt-stress and recovery conditions in 'Y1805' (Table S4). Therefore, these DEPs might simultaneously participate in the responses to salt stress and recovery. Moreover, these DEPs showed the same expression patterns at the two stages, with 10 DEPs being upregulated and 4 DEPs being downregulated compared with the respective corresponding controls. More specific DEPs were identified after recovery than under salt-stress conditions in 'Y1805'. Thus, salt stress and recovery regulated different types of proteins, allowing plants to adapt to environmental stress or growth restoration, respectively.

15 DEPs closely related to salt stress were selected from 'Y1805' for the qRT-PCR analysis (Table S1). The correlation coefficient of the relative expression levels between DEPs and qRT-PCR was significant (r = 0.802, p = 1.97E$^{-04}$). The up- or downregulation of the proteins, as determined by the proteomics analysis, was corroborated by the qRT-PCR experiment (Fig. 2C). These data demonstrated that the proteomics accurately reflected the responses of *Tritipyrum* to salt stress and recovery.

## Enrichment analysis

### DEP enrichment analysis under salt-stress conditions

GO and KEGG enrichment analyses of the common DEPs in 'Y1805' and 'CS' were performed under salt-stress conditions (Fig. S1). 'Catalytic activity' (GO:0003824), 'binding' (GO:0005488) (molecular functions), 'cell' (GO:0005623 ), 'cell part' (GO:0044464), 'membrane part' (GO:0044425), 'membrane' (GO:0016020), 'organelle' (GO:0043226) (cell components), 'cellular process' (GO:0009987), 'metabolic process' (GO:0008152) and 'cellular component organization or biogenesis' (GO:0071840) (biological process) were significantly enriched in the GO analysis. The KEGG pathways 'environmental adaptation' (ko04626), 'global and overview maps' (ko01100), 'carbohydrate metabolism' (ko00052), 'amino acid metabolism' (ko00360), 'biosynthesis of other secondary metabolites' (ko00940), 'nucleotide metabolism' (ko00230), 'signal transduction' (ko04011), 'translation' (ko03010) and 'transport and catabolism' (ko04144) were significantly enriched. These GO terms and KEGG pathways had fundamental roles in the responses to salt stress in 'Y1805' and 'CS'.

Plant antioxidant proteins are part of defense systems and play important roles in response to salt stress. 'Antioxidant activity' (GO:0004601) and 'molecular function regulator' (GO:0004867) were two 'Y1805'-specific GO terms (Fig. 3A, Figs. S1, S2). The antioxidative activity was regulated by three peroxidase (POD) superfamily proteins: peroxidase 5 AT1G14550 (TraesCS1B01G096300.1), peroxidase 47 AT4G33420 (TraesCS2B01G616700.1) and peroxidase 50-like AT4G37520 (TraesCS2A01G502100.1) (Table S5). POD plays pivotal roles in the removal of $H_2O_2$. The molecular functional regulatory protein was subtilisin chymotrypsin inhibitor-2A EMT04243 (TraesCS3A01G093900.1). The subcellular localization of 'Y1805'-specific DEPs under salt-stress conditions revealed that most were cytosolic proteins, followed by chloroplastic and then extracellular (Fig. 3C).

### DEP enrichment analysis during the recovery process

The molecular functions of common DEPs in 'Y1805' and 'CS' were 'binding', 'catalytic activity', 'antioxidant activity' and 'transporter activity' (GO:0005215) during the recovery process (Fig. S3A). In the cell component category, they were mainly 'cell', 'cell part', 'organelle' and 'extracellular region' (GO:0005576). In the biological process category, 'cellular process' and 'metabolic process' predominated. The enriched KEGG pathways were 'global and overview maps', 'amino acid metabolism', 'biosynthesis of other secondary metabolites', 'carbohydrate metabolism', 'metabolism of other amino acids' (ko00410), 'transport and catabolism', 'signal transduction', 'lipid metabolism' (ko00561) and 'energy metabolism' (ko00920) (Fig. S3B). These GO terms and KEGG pathways had major roles in the growth recovery of 'Y1805' and 'CS' after salt stress.

'Y1805' was enriched for two specific pathways, 'virion' (GO:0005634) and 'virion part' (GO:0019013), during the recovery process (Fig. 4A, Fig. S4A). Their DEPs were H/ACA ribonucleoprotein complex subunit 3-like NOP10 (TraesCS6B01G254000.1) (Table S5), which is necessary for ribosomal biogenesis and a part of the complex that catalyzes

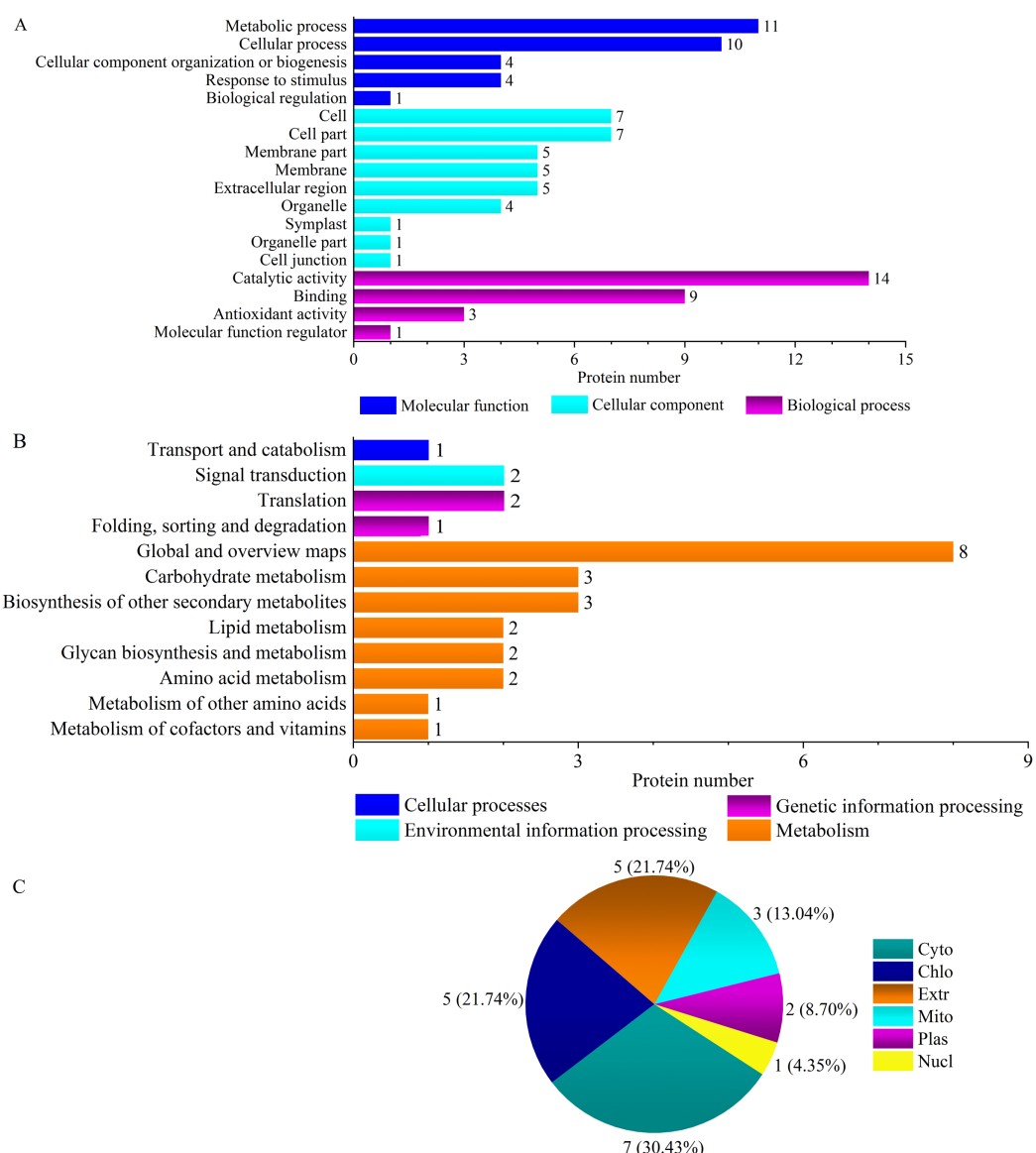

**Figure 3 GO terms (A), KEGG pathways (B) and subcellular localization (C) of 'Y1805'-specific differentially expressed proteins under salt stress.** Cyto, chlo, extr, mito, plas and nucl indicate cytosol, chloroplast, extracellular, mitochondria, plasma membrane, and nucleus, respectively.

the pseudouridylation of rRNA. Most 'Y1805'-specific DEPs localized to chloroplasts, followed by cytosol and then nuclei (Fig. 4C).

## DEP interaction analysis

PPI analysis was conducted for 'Y1805' DEPs during the salt-stress and recovery stages (Fig. 5). In total, seven nodal proteins were detected among the 44 DEPs under salt-stress conditions (Table S6). Probable cellulose synthase A catalytic subunit 8 IRX1 (TraesCS2A01G102600.1) interacted with both phenylalanine ammonia-lyase 1 PAL1 (TraesCS2D01G377600.1) and probable cellulose synthase A catalytic subunit 1 CESA1

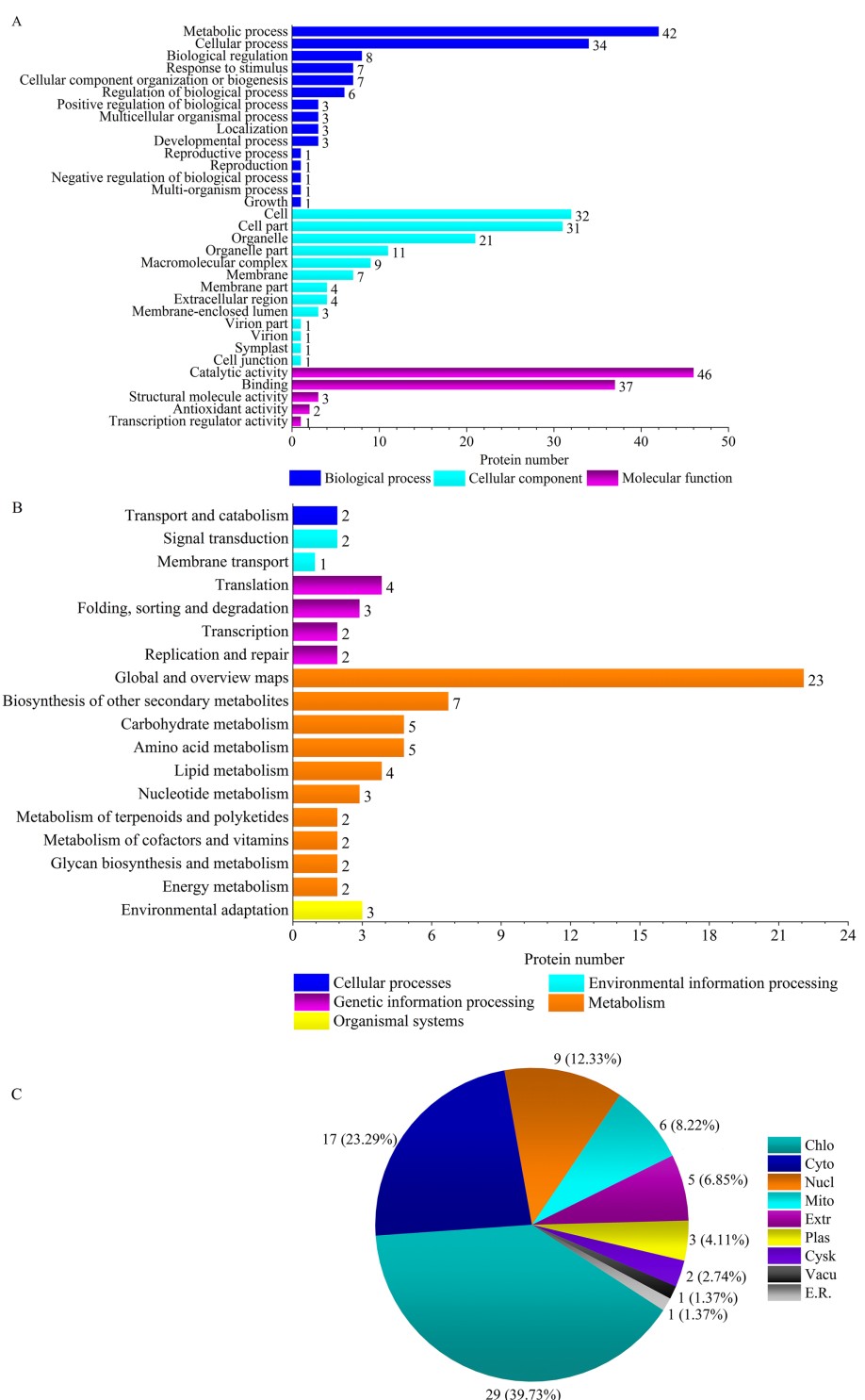

**Figure 4 GO terms (A), KEGG pathways (B) and subcellular localizations (C) of 'Y1805'-specific differentially expressed proteins during the recovery process.** Chlo, cyto, nucl, mito, extr, plas, cysk, vacu and E.R. indicate chloroplast, cytosol, nucleus, mitochondria, extracellular, plasma membrane, cytoskeleton, vacuolar membrane and endoplasmic reticulum, respectively.

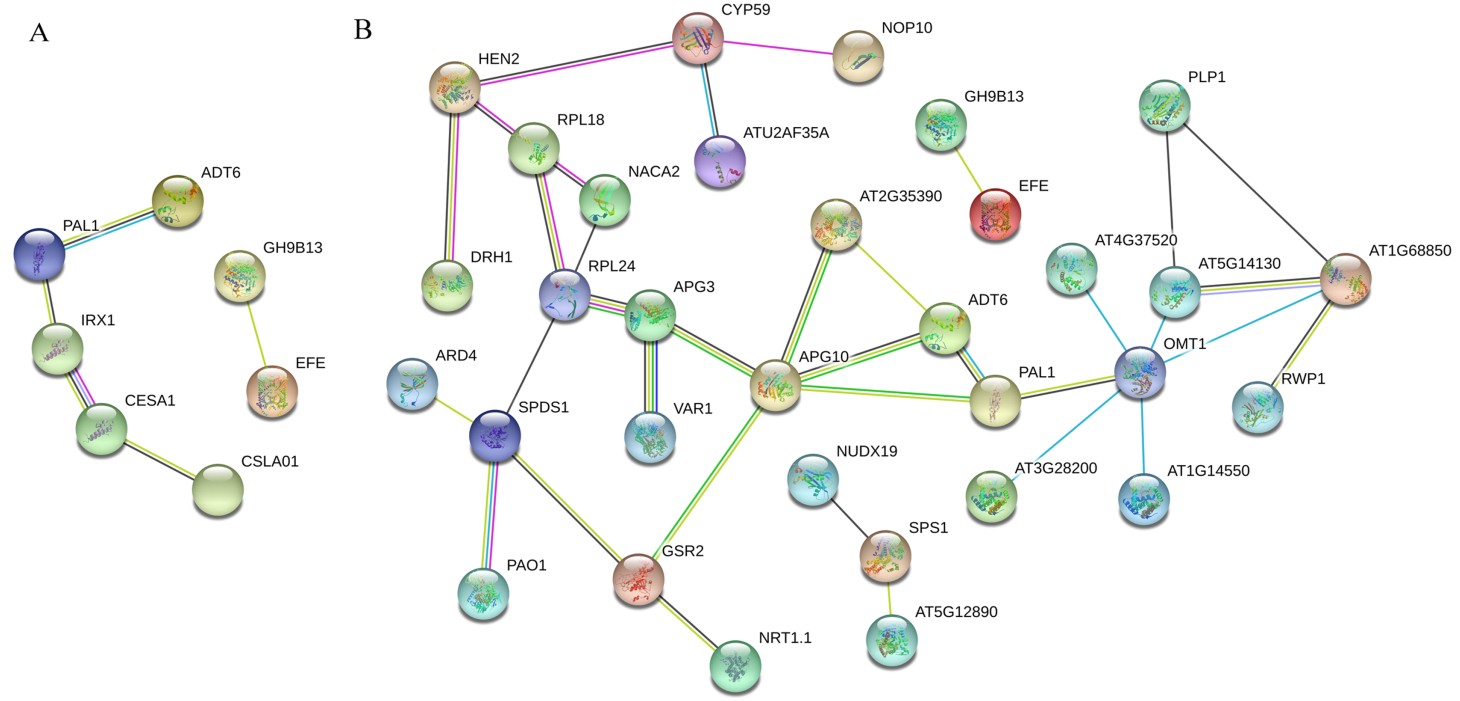

**Figure 5 *Tritipyrum* 'Y1805' network diagram of protein interactions under salt-stress (A) and recovery (B) conditions.** Yellow lines represent text mining evidence, purple lines represent experimental evidence, sky blue lines represent database evidence, black lines represent co-expression evidence, green lines represent gene neighborhoods, red lines represent gene fusions and blue lines represent gene co-occurrence database evidence. PAL1, phenylalanine ammonia-lyase 1; ADT6, arogenate dehydratase/prephenate dehydratase 6; IRX1, probable cellulose synthase A catalytic subunit 8 (UDP-forming); CESA1, probable cellulose synthase A catalytic subunit 1 (UDP-forming); CSLA01, glucomannan 4-beta-mannosyl-transferase 1; GH9B13, endoglucanase 17; APG3, peptide chain release factor APG3; APG10, (1-(5-phosphoribosyl)-5- [(5-phosphoribosylamino) methylideneamino]) imidazole-4-carboxamide isomerase; GSR2, cytosolic glutamine synthetase; AT2G35390, ribose-phosphate pyrophosphokinase 1; OMT1, flavone 3″-O-methyltransferase 1; ATG37520, peroxidase 50-like; AT5G13430, cytochrome b-c1 complex subunit rieske, mitochondrial isoform X1; AT1G68850, peroxidase 11; AT1G14550, peroxidase 5-like; AT3G28200, peroxidase 31-like; RPL18, 60S ribosomal protein L18-2-like; NACA2, nascent polypeptide-associated complex subunit alpha-like protein 2; SPDS1, spermidine synthase.

(TraesCS1B01G136200.1) (Table S5). PAL1 interacted with arogenate dehydratase/prephenate dehydratase 6 ADT6 (TraesCS2A01G292000.1), CESA1 interacted with glucomannan 4-beta-mannosyltransferase 1 CSLA01 (TraesCS6A01G169200.1), and endoglucanase 17 GH9B13 (TraesCS7A01G204500.1) interacted with ethylene-forming enzyme EFE (TraesCS1A01G089500.1). ADT6 and PAL1 are related to lignin synthesis, while IRX1 and CESA1 are involved in cellulose synthesis. The mannose polysaccharide produced by CSLA01 is a cell-wall component. Therefore, PAL1, ADT6, IRX1, CESA1 and CSLA01 were all closely related to the synthesis of the cell wall, indicating that cell-wall integrity under salt-stress conditions was crucial for salt tolerance.

In total, 32 nodal proteins were found among the 102 DEPs identified during the recovery process (Table S6). 1-(5-phosphoribosyl)-5- [(5-phosphoribosylamino) methylideneamino] imidazole-4-carboxamide isomerase APG10 (TraesCS1A01G251000.1) was the core protein in this PPI network, and it interacted with peptide chain release factor APG3 (TraesCS1B01G098300.2), cytosolic glutamine synthetase GSR2 (TraesCS4A01G266900.1), PAL1, ADT6 and ribose-phosphate pyrophosphokinase 1 AT2G35390

**Table 1 Differentially expressed protein number, GO terms and KEGG pathways identified using a proteome-transcriptome-associated analysis under salt-stress and recovery conditions in *Tritipyrum* 'Y1805'.**

| GO/pathway ID | GO terms/pathways, level 2 | Number of DEPs/ DEGs | |
| --- | --- | --- | --- |
| | | Salt stress | Recovery |
| GO:0009611 | Response to stimulus | 3 | 1 |
| GO:0004601 | Antioxidant activity | 1 | 1 |
| ko04144 | Transport and catabolism | 2 | 1 |
| ko04075 | Signal transduction | 1 | 2 |
| ko00360 | Amino acid metabolism | 2 | 4 |
| ko00940 | Biosynthesis of other secondary metabolites | 2 | 3 |
| ko00052 | Carbohydrate metabolism | 1 | 3 |
| ko00920 | Energy metabolism | 0 | 1 |
| ko00561 | Lipid metabolism | 0 | 2 |

(TraesCS6A01G088200.2). Flavor 3'-O-methylransferase 1 OMT1 (TraesCS4B01G352400.1) catalyzes the methylation of lignin precursor monoglyceride and also participates in the biosynthesis of melatonin. It interacted with AT4G37520, cytochrome b-c1 complex subunit rieske, mitochondrial isoform X1 AT5G14130 (TraesCS7A01G262000.1), peroxidase 11 AT1G68850 (TraesCS7A01G211200.1), peroxidase 5-like AT1G14550 (TraesCS1B01G096900.1), peroxidase 31-like AT3G28200 (TraesCS1B01G220600.1) and PAL1. RPL24 (TraesCS7A01G450600.1) plays a key role in the biosynthesis of thylakoid membrane proteins encoded by chloroplast genes, and it interacted with 60S ribosomal protein L18-2-like RPL18 (TraesCS1D01G105500.1), nascent polypeptide-associated complex subunit alpha-like protein 2 NACA2 (TraesCS4B01G347400.1), APG3 and spermidine synthase SPDS1 (TraesCS7B01G232700.1). OMT1 and RPL24 were the second most important hub proteins in this PPI network.

## Proteome-transcriptome-associated analysis

Using a proteome-transcriptome-associated analysis, 'response to stimulus' (GO:0009611), 'antioxidant activity', 'carbohydrate metabolism', 'amino acid metabolism', 'signal transduction', 'transport and catabolism' and 'biosynthesis of other secondary metabolites' were identified under both salt-stress and recovery conditions in 'Y1805' (Table 1). In addition, 'energy metabolism' and 'lipid metabolism' were recovery-specific pathways. In total, 13 and 25 upregulated DEPs/DEGs were identified under salt-stress and recovery conditions in 'Y1805', respectively (Table S7). Furthermore, seven DEPs/DEGs appeared under both conditions, and were probable carboxylesterase 15 CXE15 (TraesCS1D01G256800.1), abscisic acid (ABA)-inducible protein PHV A1-like HVA1 (TraesCS1D01G369800.1), ADT6, PAL1 and three uncharacterized proteins (TraesCS1A01G295800.1, TraesCS1B01G304800.1 and TraesCS2D01G417100.2) (Table S7). In addition to the common proteins, there were 6 and 18 'Y1805'-specific DEPs/DEGs identified under salt-stress and recovery conditions, respectively (Table S7).

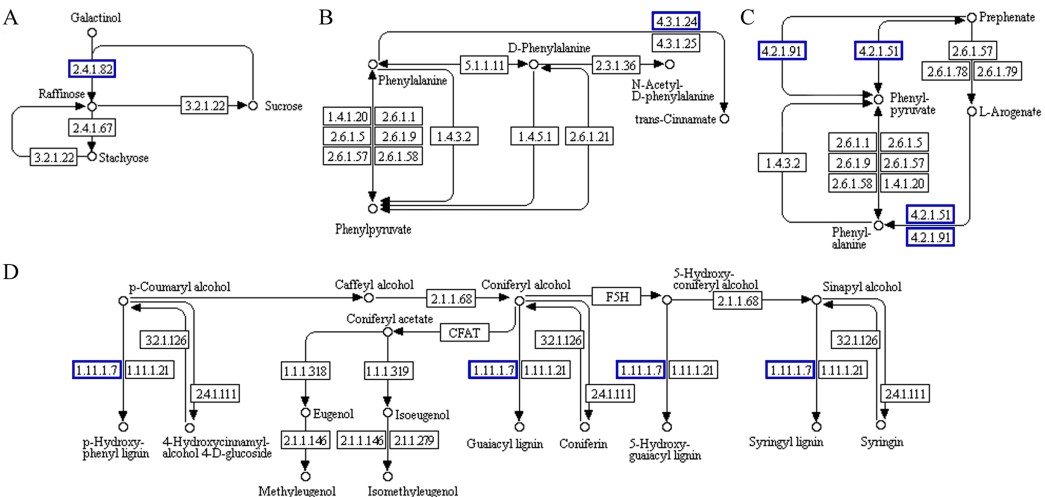

**Figure 6 Functional association diagram of expression patterns of proteins and genes under salt-stress conditions in *Tritipyrum* 'Y1805'.** (A) A part of 'galactose metabolism' pathway; (B) a part of 'phenylalanine metabolism' pathway; (C) a part of 'phenylalanine, tyrosine and tryptophan biosynthesis' pathway; (D) a part of 'phenylpropanoid biosynthesis' pathway. The blue boxes indicate that the expression patterns of the protein and gene are the same.

Compared with 'CS', four 'Y1805'-specific DEPs (AT1G14550, AT4G33420, AT4G37520, EMT04243) were found under salt-stress conditions. Among them, the antioxidant protein AT1G14550 and molecular functional regulatory protein EMT04243 displayed the same expression pattern at the transcriptional and proteome levels. Putative calcium-binding protein CaM-like (CML) 16 (TraesCS3B01G045500.1) was upregulated during the recovery process only in 'Y1805' at both the transcriptome and proteome levels (Table S7).

The functional association diagram of 'Y1805' DEPs/DEGs under salt-stress conditions is shown in Fig. 6. Among the associated proteins, galactinol-sucrose galactosyltransferase RFS1 (TraesCS3A01G092800.1) was upregulated in 'galactose metabolism' (ko00052) and is a key protein in raffinose synthesis. PAL1 and AT1G14550, which are involved in the synthesis of cinnamic acid and lignins, respectively, were upregulated in 'phenylpropanoid biosynthesis' (ko00360). In addition, PAL1 is also a key protein in 'phenylalanine metabolism'. ADT6, which takes part in phenylalanine synthesis, was upregulated in 'phenylalanine, tyrosine and tryptophan biosynthesis' (ko00400).

Probable serine acetyltransferase 1 SAT1 (TraesCS3A01G287400.1) and 1-aminocyclopropane-1-carboxylate oxidase 1 EFE (TraesCS5A01G234300.1) were upregulated in 'cysteine and methionine metabolism' (ko00270) during the recovery process in 'Y1805' (Fig. 7). They participate in the synthesis of acetyl-L-serine and ethylene, respectively. Xylanase inhibitor Xip-R1 (TraesCS4A01G173800.1), which is involved in the synthesis of chitobiose and GlcNAc, was upregulated in 'amino sugar and nucleotide sugar metabolism' (ko00520). PAL1, AT1G68850 and berberine bridge enzyme-like 27 AT5G44410 (TraesCS7B01G273600.1) were upregulated in 'phenylpropanoid biosynthesis'.

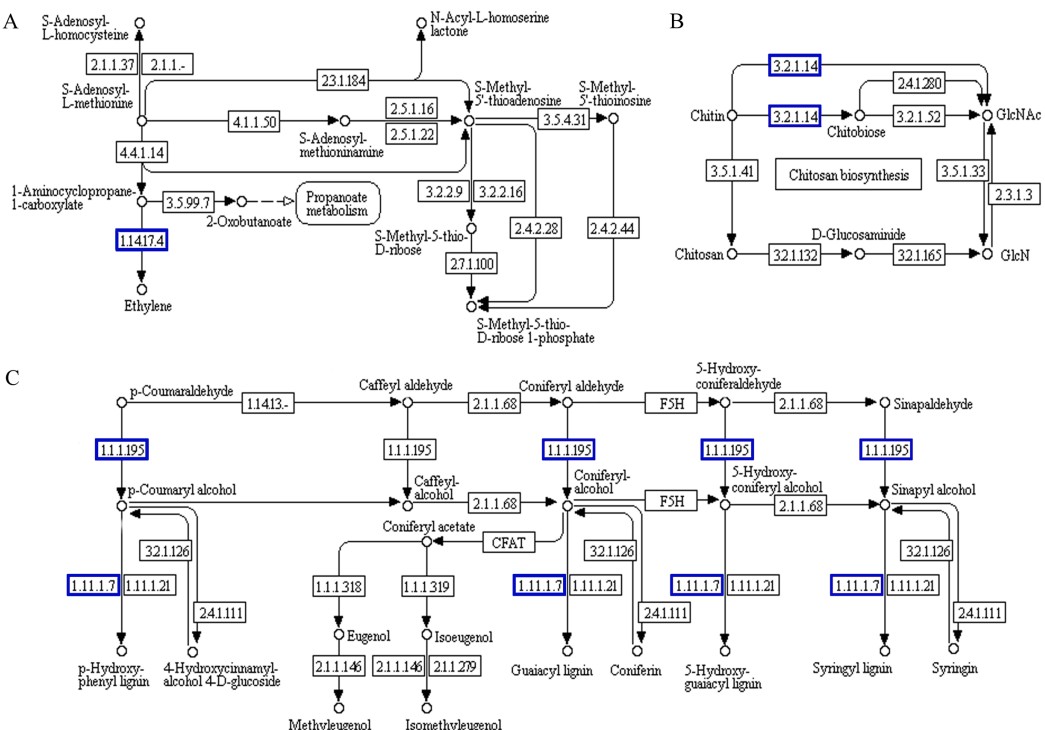

**Figure 7 Functional association diagram of expression patterns of proteins and genes during the recovery process in *Tritipyrum* 'Y1805'.** (A) A part of 'cysteine and methionine metabolism' pathway; (B) a part of 'amino sugar and nucleotide sugar metabolism' pathway; (C) a part of 'phenylpropanoid biosynthesis' pathway. The blue boxes indicate that the expression patterns of the protein and gene are the same.

## DISCUSSION

Plants, being sessile, have evolved specific and complex mechanisms in response to different abiotic stresses at the transcriptome, proteome and physiological levels. The responses of salt-tolerant common wheat 'Jing411' and salt-sensitive wheat 'CS' seedling roots to salt stress have been investigated using a proteomics analysis and 52 and 47 specific DEPs were detected, respectively (*Guo et al., 2012*). In total, 38 DEPs have been identified under salt-stress conditions in durum wheat (*Caruso et al., 2008*). In this study, the DEP quantitative analysis was carried out under salt-stress and recovery conditions in 'Y1805' and 'CS'. In total, 44 and 69 DEPs were identified in 'Y1805' and 'CS', respectively, among which 21 proteins with the same expression patterns were common to both genotypes. They represented the core genes associated with responses to salt stress. 'Y1805' had 23 specific DEPs under salt-stress conditions that should play key roles in its strong salt tolerance. Probable ATP-dependent DNA helicase CHR12 (TraesCS1A01G087900.1), HVA1, EFE, TPP3, PAL1, ADT6 and RFS1 were novel proteins identified under salt-stress conditions in 'Y1805' when compared with salt-tolerant common wheat (*Guo et al., 2012*), durum wheat (*Caruso et al., 2008*) and barley (*Mostek et al., 2015*). ATCHR12 plays an important role in the regulation of stress-induced transient growth arrest in *Arabidopsis thaliana* and allows plants to stop growth for a short

time under stress conditions (*Zhu, 2001*; *Achard et al., 2006*). Slower growth may allow plants to redirect resources to overcome or temporarily cope with stress. Here, CHR12 was 'Y1805'-specific under salt-stress conditions, during which the expression level was very high (4.0572), but not after recovery.

After recovery, 73 'Y1805'-specific DEPs were identified that should contribute to its fast recovery after removing the salt stress. In total, 14 DEPs were identified under both salt-stress and recovery conditions in 'Y1805'. More specific DEPs were identified after recovery than during the salt stress in 'Y1805'. Therefore, the responses to salt stress and recovery were two independent, but closely related, processes.

## Protein responses to salt stress

### Antioxidant proteins

As members of the defense system, plant antioxidant proteins remove ROS and reduce the damage caused by oxidative stress. The overexpression of class III peroxidase genes enhances the antioxidant capacity of maize under abiotic stress conditions (*Wang et al., 2015*). POD (B4G0X5) was upregulated in both salt-tolerant and salt-sensitive maize to improve the salt tolerance of the two maize varieties (*Chen et al., 2019*). Here, the 'antioxidant activity' pathway was enriched, and three peroxide superfamily proteins (AT1G14550, AT4G33420 and AT4G37520) were identified under salt-stress conditions only in 'Y1805'. AT1G14550, which could remove excessive $H_2O_2$, thereby enhancing the antioxidant capacity, was upregulated at both the transcriptional and protein levels. However, no POD was found in 'CS', resulting in excessive ROS accumulation, which led to oxidative damage.

### Osmoregulation

Soluble sugars participate in osmotic regulation and also play important roles in maintaining protein stability. Glycosyltransferase (A0a1d6ewj3) was upregulated in both salt-tolerant and -sensitive maize cultivars, and plays an active role in salt tolerance (*Chen et al., 2019*). RFS1 is involved in the synthesis of cottonseed sugar, which acts as an osmotic protective agent in plant cells (*Nishizawa, Yabuta & Shigeoka, 2008*). In this work, 'carbohydrate metabolism' was significantly enriched. Trehalose has an important physiological role in osmotic protection, and it improves the tolerance of plants to abiotic stress (*Wang et al., 2020*). In this experiment, upregulated trehalose-phosphate phosphatase 3 TPP3 (TraesCS2A01G161200.1) was detected only in 'Y1805', and it could remove phosphate from trehalose-6-phosphate to produce free trehalose. Therefore, soluble sugars might contribute to the strong salt tolerance of 'Y1805'.

### Phytohormones

ABA plays an important role in regulating water balance, osmotic protection and cell wall component synthesis, and it is also involved in the control of phosphatidyl inositol signaling and chromatin-mediated stress tolerance mechanisms, which alleviate the physiological effects of salt stress and enhance the protective mechanisms against stress (*Szypulska, Jankowski & Weidner, 2017*). Low doses of ABA (10–100 μM) induce
antioxidant defense mechanisms to prevent oxidative damage to membrane lipids and proteins (*Jiang & Zhang, 2001*; *Wang, Kuang & He, 2010*). Pathogenic-associated protein 10 is involved in ABA synthesis, which is upregulated in salt-tolerant maize '8723' but stable in salt-sensitive maize 'P138' (*Chen et al., 2019*). A similar result was obtained in our study. HVA1 was upregulated at both the transcriptional and protein levels under salt-stress conditions only in 'Y1805'. The upregulated expression of HVA1 produces ABA, which is beneficial to maintaining the water balance and preventing oxidative damage.

Ethylene induces the occurrence of adventitious roots and root hairs. The presence of root hairs increases the absorption area of roots, and they secrete a variety of substances, such as organic acids. Moreover, ethylene increases the flavonol content of guard cells. Flavonols act as antioxidants, reduce ROS levels in guard cells and inhibit stomatal closure (*Murata, Mori & Munemasa, 2015*). An increase in 1-aminocyclopropane-1-carboxylic acid oxidase leads to a large amount of ethylene production (*Qin et al., 2007*). An ethylene receptor (a negative regulator of ethylene signaling) expression level increased more than nine-fold in salt-sensitive wheat 'Jinan 177' and only two-fold in salt-tolerant 'Shanrong No. 3' in response to salt stress. Therefore, the lower ethylene receptor expression in 'Shanrong No. 3' under salt-stress conditions may be partly responsible for its lower sensitivity to salinity than 'Jinan 177' (*O'Malley et al., 2005*; *Wang, Kuang & He, 2010*). Here, upregulated EFE was found only in 'Y1805' under salt-stress conditions and is beneficial for its strong salt tolerance. Under salt-stress conditions, upregulated HVA1 and EFE play vital roles in the salt tolerance of 'Y1805'.

## Protein responses to recovery

### Cell wall synthesis

UDP-glucose dehydrogenase (UGD) plays a key role in the nucleotide sugar biosynthetic pathway, because its product, UDP-glucuronic acid, is the common precursor for arabinose, xylose, galacturonic acid and apiose residues found in cell walls, which play important roles in primary cell-wall formation (*Reboul et al., 2011*; *Siddique et al., 2012*). Here, an upregulated UDP-glucose 6-dehydrogenase 2, UGD2 (TraesCS4A01G237900.1), which improves the synthesis of cell-wall polymers, was detected during the recovery process only in 'Y1805'. Caffeic acid 3-O-methyltransferase (OMT) catalyzes the multi-step methylation reactions of hydroxylated monomeric lignin precursors, which have important roles in lignin biosynthesis (*Ma & Xu, 2008*). OMT1 was a 'Y1805'-specific upregulated protein, and it is involved in lignin biosynthesis. With the increase in lignin synthesis, 'Y1805' was able to repair cell walls rapidly during the recovery process, resulting in a rapid return to normal growth. These upregulated UGD2 and OMT1 proteins in 'Y1805' strengthen cell walls and play master roles in the line growth recovery.

### Improved respiration

Stress conditions stimulate the expression of GPDHC1, which is related to the availability of oxygen and a key protein affecting mitochondrial respiration (*Shen et al., 2006*). In this study, GPDHC1 (TraesCS3A01G330600.1) was upregulated only in 'Y1805' during the

recovery process, indicating that its respiration was improved, which would provide essential energy and intermediate metabolites for the plant's rapid recovery after being subjected to salt stress.

## Phytohormones and signal transduction

During the recovery process, HVA1 was upregulated in both 'Y1805' and 'CS', but the expression level in 'Y1805' was higher than that in 'CS' (Table S6). OMT1 catalyzes the transfer of a methyl group onto N-acetylserotonin, which produces melatonin (N-acetyl-5-methoxytryptamine) (*Byeon et al., 2015*). Melatonin has antioxidative effects in plants. It scavenges ROS mainly by providing electrons. Melatonin also promotes the formation of lateral roots. Indole-3-acetic acid (IAA)-amino acid hydrolase ILR1-like 2 produces free IAA by hydrolyzing IAA-amido (*Leclere et al., 2002*; *Rampey et al., 2004*). This enzyme (TraesCS3B01G283900.1) was upregulated only in 'Y1805', resulting in the production of free IAA, which promotes plant growth. Salicylic acid (SA) reduces water loss by inhibiting stomatal opening to alleviate salt stress. EPS1 (a BAHD acyltransferase-family protein) is an essential protein for pathogen-induced SA accumulation (*Torrens-Spence et al., 2019*). Here, EPS1 (TraesCS3A01G367900.1) was upregulated only in 'Y1805', which resulted in an SA accumulation. Together, these phytohormones had a critical impact on the rapid growth recovery of 'Y1805'.

The responses of plants to abiotic stress are to consolidate their growth and development by regulating complex signaling networks for stress adaptation (*Abreu et al., 2013*). In plants, $Ca^{2+}$ signals play important roles in response to abiotic stresses, such as salinity, drought and temperature. The CML proteins ($Ca^{2+}$ sensors) play unique roles in plant $Ca^{2+}$-signaling pathways (*Ogunrinde et al., 2017*). The $Ca^{2+}$ signaling network is closely related to the activation of the salt overly sensitive 'signal transduction' pathway that regulates cellular $Na^+$ and $K^+$ homeostasis in Arabidopsis (*Zhu, 2002*; *Brini et al., 2007*). In this study, the 'signal transduction' pathway was significantly enriched, and CML16 was upregulated only in 'Y1805' after recovery. Therefore, $Ca^{2+}$ signaling was involved in the responses to recovery in 'Y1805'.

## Transcriptional regulation and error information processing

DExH-box ATP-dependent RNA helicase DExH10 (HEN2) assists the exosome-mediated processing and/or degradation of small nucleolar RNAs and their precursors, microRNA precursors and long non-coding RNAs, as well as a large number of spurious transcripts derived from antisense and non-annotated regions. In addition, HEN2 is involved in the degradation of excised introns and incompletely spliced or otherwise mis-transcribed or mis-processed mRNAs (*Lange et al., 2014*). DEAD-box ATP-dependent RNA helicase 14 (DRH1) is a novel type of ATP/dATP-dependent RNA helicase and polynucleotide-dependent ATPase that is capable of unwinding double-stranded RNA in the presence of ATP or dATP and of hydrolyzing ATP (*Okanami, Meshi & Iwabuchi, 1998*). In this study, 'transcription regulation activity' pathway was enriched, and upregulated HEN2 (TraesCS2A01G154400.1) and DRH1

(TraesCS3A01G130200.1) were detected only in 'Y1805' during the recovery process, indicating that transcriptional regulation and error information processing performed crucial roles in the line rapid recovery from salt stress.

The wheat DEPs were enriched in 'carbon metabolism', 'detoxification', 'defense', 'signal transduction' and 'companionship' under salt-stress conditions as determined by a proteomics analysis (*Caruso et al., 2008*; *Wang et al., 2008*; *Guo et al., 2012*; *Ma et al., 2020*). Here, 'carbohydrate metabolism', 'antioxidant activity' and 'environmental adaptation' and 'signal transduction' were also enriched in 'Y1805'. What's more, the salt tolerance of 'Y1805' might be attributed to cell-wall strengthening, transient growth arrest, osmoregulation and ABA regulation.

## DEP interactions

Cell wall integrity is an important factor that determines salt-stress tolerance and plant growth (*Feng et al., 2018*; *Zhao et al., 2018*). Lignin is produced and the cell wall is hardened through the actions of ADT6 and PAL1 (*Huang et al., 2010*). CEAS1 is involved in the formation of the primary cell wall. IRX1 participates in the formation of the secondary cell wall and plays an important role in the thickening of the xylem cell wall (*Turner & Somerville, 1997*). CSLA01 is involved in the synthesis of galactomannan, which is a non-cellulose polysaccharide in plant cell walls. ADT6, PAL1, CEAS1, IRX1 and CSLA01 interacted with each other under salt-stress conditions in 'Y1805', and they were closely related to the synthesis and strength of the cell wall. In addition, IRX1 and CSLA01 were 'Y1805'-specific DEGs. Therefore, the cell wall performed major roles in the salt tolerance of 'Y1805'.

Abiotic stresses, such as salinity, drought, osmotic, cold and freezing temperatures produce cellular water deficits, which lead to the accumulation of highly hydrophilic LEA proteins (*Battaglia et al., 2008*; *Hundertmark & Hincha, 2008*). Nine LEA DEGs (*Tel3E01G270600, Tel3E01G426400, Tel3E01G628300, Tel4E01G475800, Tel5E01G589800, Tel5E01G665500, Tel6E01G622800, Tel6E01G623000* and *Tel7E01G989400*) originating from *Thinopyrum elongatum* were detected among top 30 DEGs under salt-stress conditions in 'Y1805' according to our transcriptome analysis. Histidine is a key component of LEA proteins. APG10 is not only a hydrophilic protein, but also an important protein in histidine biosynthesis. In this study, APG10 was the hub protein during the recovery process in 'Y1805', and it was downregulated. The result revealed that LEA proteins contributed to salt tolerance, and their expression levels fell rapidly during the recovery process. OMT1 is involved in the biosynthesis of lignin, which is used to repair the cell wall rapidly during the recovery process in 'Y1805'. It also produced melatonin, which improved the rapid growth recovery of 'Y1805'. RPL24 is an important protein for photosynthesis and optimal plastid performance, and it plays a key role in the biosynthesis of thylakoid membrane proteins encoded by chloroplast genes (*Romani et al., 2012*). OMT1 and RPL24 were the second most important hub proteins in the 'Y1805' PPI network. They were both 'Y1805'-specific and upregulated during the recovery process.

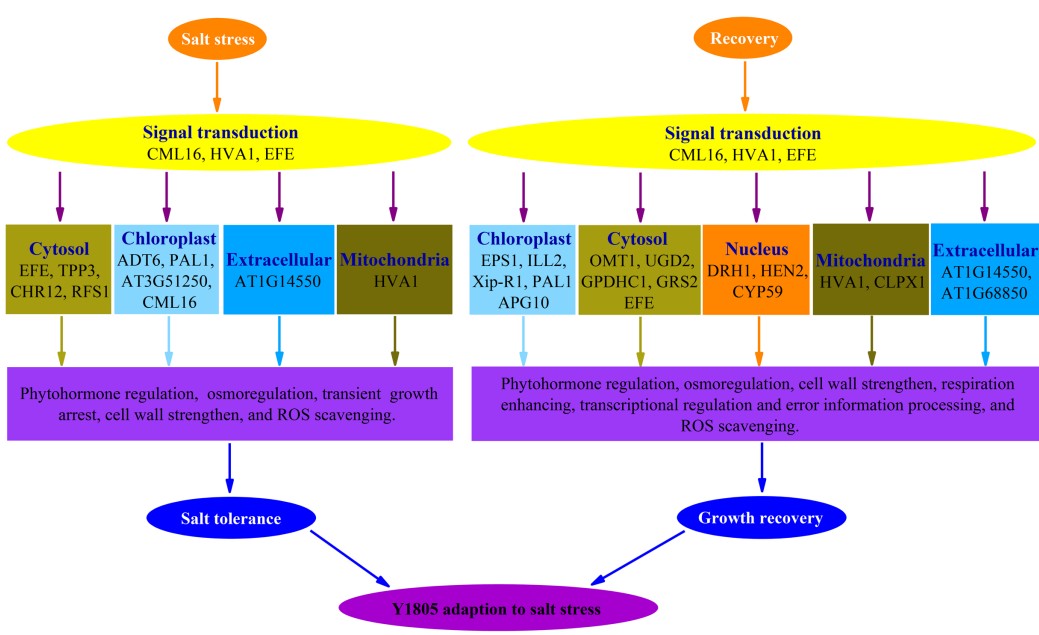

**Figure 8 Schematic representation of the responses to salt stress and recovery in *Tritipyrum* 'Y1805'.**

# CONCLUSION

Forty-four and 102 DEPs were identified in 'Y1805' under salt-stress and recovery conditions, respectively, using the DIA proteomics techniques. In total, 13 and 25 upregulated DEPs/DEGs were found under salt-stress and recovery conditions, respectively, using a proteome-transcriptome-associated analysis. Compared with 'CS', 'Y1805' had specific responses to salt stress and recovery. Scavenging ROS, osmoregulation, phytohormone regulation, transient growth arrest and cell-wall strengthening might be of paramount importance in its strong salt tolerance (Fig. 8). After recovery, 'Y1805' restored growth by cell-wall strengthening, respiration enhancement, phytohormone regulation, signal transduction, and transcriptional regulation and error information processing (Fig. 8). This work provides a new perspective on the molecular mechanisms underlying the adaptation of *Tritipyrum* to salt stress, and the results may accelerate salt-tolerant crop breeding programs.

# ACKNOWLEDGEMENTS

Special thanks are due to Pr. Adam J. Lukaszewski (University of California Riverside, USA) for the technical assistance in the experiments. We thank International Science Editing for editing this manuscript.

## Funding

This work was supported by the National Natural Science Foundation of China (31860380, 32160442 and 31660390), and the Science Foundation of Guizhou Province [(2018)5781

and (2019)1110]. The funders had no role in study design, data collection and analysis, decision to publish, or preparation of the manuscript.

## Grant Disclosures
The following grant information was disclosed by the authors:
National Natural Science Foundation of China: 31860380, 32160442 and 31660390.
Science Foundation of Guizhou Province: [(2018)5781 and (2019)1110].

## Competing Interests
The authors declare that they have no competing interests.

## Author Contributions
- Rui Yang performed the experiments, analyzed the data, prepared figures and/or tables, authored or reviewed drafts of the paper, and approved the final draft.
- Zhifen Yang performed the experiments, prepared figures and/or tables, authored or reviewed drafts of the paper, and approved the final draft.
- Ze Peng performed the experiments, prepared figures and/or tables, and approved the final draft.
- Fang He conceived and designed the experiments, analyzed the data, prepared figures and/or tables, and approved the final draft.
- Luxi Shi performed the experiments, prepared figures and/or tables, and approved the final draft.
- Yabing Dong performed the experiments, prepared figures and/or tables, and approved the final draft.
- Mingjian Ren conceived and designed the experiments, authored or reviewed drafts of the paper, and approved the final draft.
- Qingqin Zhang conceived and designed the experiments, authored or reviewed drafts of the paper, and approved the final draft.
- Guangdong Geng conceived and designed the experiments, authored or reviewed drafts of the paper, and approved the final draft.
- Suqin Zhang conceived and designed the experiments, authored or reviewed drafts of the paper, and approved the final draft.

## Data Availability
The mass spectrometry proteomics data are available at the ProteomeXchange Consortium *via* the PRIDE partner repository: PXD026671.

## Supplemental Information
Supplemental information for this article can be found online at http://dx.doi.org/10.7717/peerj.12683#supplemental-information.

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
