# Peer review of "Integrated transcriptomic and proteomic analysis of Tritipyrum provides insights into the molecular basis of salt tolerance"

_PeerJ, doi:10.7717/peerj.12683_

## Round 0.1 · original submission · Major Revisions

Authors should address all the comments and suggestions raised by both reviewers in their review reports. Please highlight all the changes in your revised manuscript.

Reviewer 1 ·

Basic reporting

It sounds relevant.

Experimental design

Moderately okay.

Validity of the findings

Considerable.

Additional comments

The manuscript reveals differential tolerance to salt stress through proteomics analysis supported by the downstream qPCR experiments. The work is a straightforward proteomics analysis, which was further annotated based on the existing database search. Although the manuscript may advance our understanding of salt tolerance in Tritipyrum, I would suggest some improvements to be done before considering for publication.
Line 94: How do you know that these genotypes are having contrasting tolerance to salt stress? Any reference or preliminary experiments on that?
Line 104: Although the sensitivity to NaCl depends on species/cultivars, 250 mM NaCl is too high. Why did you choose this concentration for inducing salt stress? How about the pH of the solution culture?
Line 118: Please briefly write the protein extraction method.
Line 138: List the names of the genes studied for qRT-PCR.

·

Basic reporting

The manuscript by Yang R and colleagues delineates the molecular understanding of salt tolerance in Wheat (Triticum aestivum) utilizing both transcriptomic and proteomic analysis. The authors utilized salt-tolerant and salt-sensitive wheat to find differentially expressed genes and proteins to address the molecular pathways and processes important for the salt-tolerant phenotype in wheat. I found this study to be interesting, well-crafted, and insightful.

Experimental design

1. qPCR validations were carried out using the same samples used for proteomic analysis. I would recommend using separate samples for the validation process to ensure the reproducibility of the findings

Validity of the findings

2. The authors mentioned differentially expressed proteins in other species like barley, rice, sweet potatoes (in the introduction). It would be interesting to know the common DIPs among different species to single out the most prominent molecular pathways/processed involved in salt tolerance.
3. It would be helpful to dissect the role of some of the differentially expressed genes/proteins experimentally using genetic knockdown/knockout or overexpression experiments.

Additional comments

1. I would suggest elaborating figure legends to ensure better readability.

---

## Round 0.2 · Minor Revisions

The revised manuscript is an improved version but the Section Editor notes some outstanding issues that must be addressed:

> The way in which the data is presented prevents the reader from moving forward on any information provided. GO: annotations assist in the area, and by not connecting an attachment to the individual hypothetical peptides do not allow the reader to assess the authors interpretation. In figure 2b there are assignments into 15 groups of interactions; which peptide goes with which? In figures 3 and 4 there are bar and pie charts with general assessments; to better understand the assignments which peptides were assigned to the categories. For instance if I were to pick a random Traes listing, which GO and KEGG was it assigned to and where did it fall withing the provided Venn bin? Also the text annotations of GO need to be accompanied by the proper GO: numerical assignments to streamline interpretation.

Reviewer 1 ·

Basic reporting

May be considered for publication.

Experimental design

May be considered for publication.

Validity of the findings

May be considered for publication.

·

Basic reporting

The authors successfully addressed all of my questions and concerns.

Experimental design

No comment

Validity of the findings

No comment

---

## Round 0.3 · accepted · Accept

The original Academic Editor is no longer available and so I am making a decision in my role of Section Editor.

It appears all the suggested revisions have been fulfilled; the manuscript is ready to move forward. The annotation terms added were helpful. I am hopeful that the information provided here may lead toward new insights in dealing with salt tolerance. Congratulations on your efforts.